

# Ionospheric Irregularities Reconstruction Using Multi-Source Data Fusion via Deep Learning

Penghao Tian[1], Bingkun Yu[1,2,3], Hailun Ye[1], Xianghui Xue[1,3,4], Jianfei Wu[1], and Tingdi Chen[1,3]

[1]School of Earth and Space Sciences/Deep Space Exploration Laboratory, University of Science and Technology of China, Hefei, China
[2]Institute of Deep Space Sciences, Deep Space Exploration Laboratory, Hefei, China
[3]Anhui Mengcheng Geophysics National Observation and Research Station, University of Science and Technology of China, Hefei, China
[4]Hefei National Laboratory, University of Science and Technology of China, Hefei, China

**Correspondence:** Xianghui Xue (xuexh@ustc.edu.cn); Bingkun Yu (bkyu@ustc.edu.cn)

**Abstract.** The ionospheric sporadic E (Es) layer is the intense plasma irregularities between 80 and 130 km in altitude, which is generally unpredictable. Reconstructing the morphology of sporadic E layer is not only essential for understanding the nature of ionospheric irregularities and many other atmospheric coupling systems, but also useful to solve a broad range of demands for reliable radio communication of many sectors reliant on ionosphere-dependent decision-making. Despite the efforts of many

empirical and theoretical models, a predictive algorithm with both high accuracy and high efficiency is still lacking. Here we introduce a new approach for Sporadic E Layer Forecast using Artificial Neural Networks (SELF-ANN). The prediction engine is trained by fusing observational data from multiple sources, including high-resolution ERA5 reanalysis dataset, COSMIC RO measurements, and integrated data from OMNI. The results show that the model can effectively reconstruct the morphology of the ionospheric E layer with intraseasonal variability by learning complex patterns. The model obtains good performance

and generalization capability by applying multiple evaluation criteria. The random forest algorithm used for preliminary processing shows that local time, altitude, longitude, and latitude are significantly essential for forecasting the E-layer region. Extensive evaluations based on ground-based observations demonstrate the superior utility of the model in dealing with unknown information. The presented framework will help us better understand the nature of the ionospheric irregularities, which is a fundamental challenge in upper atmospheric and ionospheric physics. Moreover, the proposed SELF-ANN can provide a

significant contribution to the development of the prediction of ionospheric irregularities in the E layer, particularly when the formation mechanisms and evolution processes of the Es layer are not well understood.

## 1 Introduction

The ionospheric E layer irregularities, or sporadic E (Es), are dense layers of metallic ions in the lower thermosphere between 80 and 130 km, a region that is characterized by complicated atmospheric dynamics and nonlinear plasma processes

(Matsushita and Reddy, 1967; Whitehead, 1970; Mathews, 1998; Haldoupis, 2011). The intense plasma irregularities within Es layers, which is representative of the complex interaction between the neutral atmosphere and the ionosphere, can cause



perturbations and scintillation in radio signals due to a large vertical gradient in electron density (Pavelyev et al., 2007; Zeng and Sokolovskiy, 2010; Sokolovskiy et al., 2014). The phenomenon has attracted considerable attention over the last decades, which led to numerous scientific studies (Arras et al., 2008; Lei et al., 2007, 2008; Chu et al., 2014; Tsai et al., 2018; Arras and
Wickert, 2018; Yu et al., 2019; Shinagawa et al., 2021; Ye et al., 2023). However, the reconstruction of the global ionospheric Es layer is a particular challenge due to the different mechanisms responsible for the formation and spatial-temporal variations of Es layers at different latitudes (Carter and Forbes, 1999; Whitehead, 1961, 1989; Raghavarao et al., 2002; Kirkwood and Nilsson, 2000; Lühr et al., 2021; MacDougall et al., 2000). Therefore, the modeling of ionospheric E-layer morphology and dynamics is essential for the near-real-time forecast and long-term prediction of the ionospheric parameters to serve the
satellite-based communication and navigation needs (Li et al., 2021).

Since the first theoretical descriptions of the ionospheric layers by Chapman (1931a, b), empirical or theoretical models have been developed by understanding the physical, chemical, and transport processes that control the variability of coupled thermosphere-ionosphere system. For instance, the International Reference Ionosphere (IRI), Thermosphere-Ionosphere-Electrodynamics General Circulation Model (TIE-GCM), Thermosphere-Ionosphere-Mesosphere-Electrodynamics General
Circulation Model (TIME-GCM), and the Ground to topside model of the Atmosphere and Ionosphere for Aeronomy (GAIA) are some of the well-established empirical or theoretical models that are widely used in ionospheric modeling (Bilitza et al., 2022; Priyadarshi, 2015; Qian et al., 2014; Roble and Ridley, 1994; Jin et al., 2011). Apart from the ionospheric numerical models mentioned above, several studies have proposed empirical and theoretical models specifically tailored for E-layer that assist in gaining a deeper insight into the evolution of the E-region morphology. Titheridge (2000) provided a refined set of
equations for modeling the peak of the ionosphere E-layer, building upon the IRI model and giving a good overall representation of the E-regions of the ionosphere. Resende et al. (2017) employed the Ionospheric E-Region Model (MIRE) to investigate the competition between tidal winds and electric fields in the formation of blanketing Es layers. Yu et al. (2022) constructed an empirical model of the Es layers using the multi-variable nonlinear least-squares-fitting method, which provides a comprehensive description of the climatology and global variation of Es layers.

The earlier studies have, to a certain extent, contributed to our comprehension of the complicated interaction between various factors that influence the formation and variability of the Es layer. However, discrepancies arise between the results of conventional models and real-world observations regarding the reproduction of Es local morphology, especially when analyzing the short-term evolution. The primary reason is the complex formation mechanism of the Es layer, which involves various physical phenomena such as gravity wave breaking in the upper atmosphere (Djuth et al., 2010; Vadas and Liu, 2009), the combined
effects of ionospheric electric fields and horizontal neutral winds (Nygren et al., 1984; Shinagawa et al., 2017; Yu et al., 2021b), intense geomagnetic activities (Thayer and Semeter, 2004; Johnson and Heelis, 2005; Pedatella, 2016), and chemical reactions of metallic ions (Plane, 2012; Wu et al., 2021). This complexity poses a significant challenge for traditional methods to capture all the underlying physical mechanics accurately. For instance, Shinagawa et al. (2021) conducted a comparison between the vertical ion convergence (VIC) by wind shear obtained from the GAIA model and the observed foEs from an ionosonde.
The correlation coefficients at 110 and 130 km altitude are only 0.35 and 0.34, respectively. It is still difficult to numerically reproduce the Es structures. Furthermore, with the continued increase in missions devoted to exploring the vicinity of Earth's



space, an extensive array of data has been amassed. Notable examples of these include Global Positioning System/Meteorology (GPS/MET), Challenging Mini-satellite Payload (CHAMP), Gravity Recovery and Climate Experiment (GRACE), and Constellation Observing System for Meteorology, Ionosphere, and Climate (FORMOSAT-3/COSMIC), among others. Regrettably,

traditional models have proven inadequate for fully harnessing this vast volume of data.

Reincarnation of artificial neural networks (ANN) in the form of deep learning has improved the accuracy of several pattern recognition tasks, such as classification of objects, scenes and various other entities in digital images (Schmidhuber, 2015; LeCun et al., 2015; Silver et al., 2017). The rapid and pervasive progress of artificial intelligence (AI) research has profoundly influenced several scientific domains, including geoscience (Reichstein et al., 2019; Ham et al., 2019; Salcedo-Sanz et al., 2020;

Yu and Ma, 2021). Examples include conditional generative adversarial networks to generate farside solar magnetograms (Kim and Cho, 2019), random forest algorithm for assessing skill in forecasting marine heatwaves (Giamalaki et al., 2022) and studying the presence of post-storm cooling in the middle-thermosphere using artificial neural networks (Licata et al., 2022), to name a few. It has been demonstrated that machine learning confers significant advantages in the field of fitting complex coupled systems, as opposed to conventional methodologies that typically rely on experimental or physical models.

Notably, while deep learning has gained increasing traction in the geosciences field, there remains a lack of research applying artificial intelligence techniques for the prediction of global Es layer morphology. Tian et al. (2022) apply the deep learning approach, which is adapted from 3D U-Net, to extract the latent correlation between the lower atmosphere and the global ionospheric Es layer. Comprehensive quantitative analysis is also provided through multiple evaluation metrics. The work has made a significant contribution to global Es layer modeling using machine learning techniques. However, limitations persist in

the study, particularly in terms of interpretability and serviceability.

Hence, we present a new deep learning model for Sporadic E Layer Forecast using Artificial Neural Networks, called SELF-ANN, to reconstruct the structure of the global Es layer by employing multi-source data fusion. The proposed algorithm is implemented by fusing observational data from multiple sources, including high-resolution ECMWF reanalysis v5 (ERA5) dataset, FORMOSAT-3/COSMIC radio occultation (RO) measurements, and integrated data from OMNI. To assess the rela-

tive importance of individual variables, such as the troposphere, stratosphere, geomagnetic and solar activities, in the prediction of the model, we employed a conventional machine learning method, i.e. random forest regression (RFR). The results indicate that the local spatiotemporal information of the E-region exhibits a more significant association with Es occurrence than other types of features. In addition, comprehensive statistical analysis reveals the correlation coefficient of 0.607 between the prediction of the model and the observation from the COSMIC RO dataset. This trained model is particularly effective for forecasting

the morphology distribution and seasonal evolution of the Es layer, whose formation mechanism and underlying processes are complex. Additional validation was conducted using the foEs data obtained from an ionosonde station situated in Beijing, China to ensure the generalizability of the SELF-ANN model. The hourly correlation coefficient between the observed foEs data and the model prediction reaches 0.531, thereby providing confidence in the practical utility of the model. We has designed and implemented graphic user interface (GUI) that integrates a well-trained model and presents a user-friendly interface. The

open-source software tool is freely available for the specific needs of researchers in the community who are interested in the Es layer. Moreover, this tool can facilitate the exploration of large-scale variations of Es layer and other climate features, as well





as the incorporation of ionosonde observations via data assimilation, resulting in a better description of the sporadic E layer distribution.

## 2 Methods

### 2.1 Data

ERA5, the latest iteration of reanalysis data provided by the European Center for Medium-Range Weather Forecasts (ECMWF), represents a significant advancement over its predecessor, ERA-Interim. Released in 2016, ERA5 offers enhanced resolution capabilities with a comprehensive coverage of 137 hybrid sigma/pressure (model) levels along the vertical axis, spanning from 1,012 hPa (0.01 km) to 0.01 hPa (80.3 km) (Hersbach et al., 2020). Notably, ERA5 incorporates a multitude of reprocessed datasets and incorporates recent instrumentation that was not integrated into ERA-Interim, supplementing traditional observation data sourced from the Global Telecommunication System. A key addition to ERA5 is the inclusion of data from the Aeolus satellite, the first global wind measurement satellite launched by the European Space Agency (ESA). This new dataset provides invaluable wind measurements, augmenting ERA5's accuracy and reliability. In this study, the reanalysis production provides the high-resolution data for lower atmosphere, which is utilized as the input information.

The FORMOSAT-3/COSMIC (FORMOsa SATellite-3/Constellation Observing System for Meteorology, Ionosphere, and Climate) constellation is comprised of six microsatellites that traverse the Earth's orbit at an initial altitude of 800 km (Rocken et al., 2000; Schreiner et al., 2007; Anthes et al., 2008). These microsatellites are tasked with performing radio occultation measurements, which allows for the collection of data on both the neutral atmosphere and ionosphere. Ionospheric scintillation of radio signals occurs when a radio wave passes through plasma density irregularities in the ionosphere (Weber et al., 1985). The S4 index is defined as the standard deviation of the detrended intensity of received signals normalized to the average signal intensity (Briggs and Parkin, 1963; Yue et al., 2014; Schreiner et al., 2011). This index has proven to be an effective measure for quantifying ionospheric scintillation, and it is widely used in research (Yue et al., 2014; Yu et al., 2019, 2020, 2021a; Arras et al., 2009; Arras and Wickert, 2018; Ye et al., 2021, 2023). Moreover, the OMNI dataset, a multi-source data collection of the near-Earth solar wind environment with hourly resolution, serves as auxiliary data for this research, i.e., it provides geomagnetic and solar activity information (King and Papitashvili, 2005; Papitashvili and King, 2020).

The objective of our research is to establish latent mapping connections between external driving factors and the ionospheric E region. The present study has opted for lower atmospheric parameters (wind, temperature, etc.) and measures of geomagnetic and solar activities, as the raw input variables. The ionospheric irregularities in the E region were represented by the maximum values of amplitude scintillation S4 index (S4max) data observed between altitudes of 80~130 km during a seven-year period spanning from 2008 to 2014. We employed linear interpolation to obtain the raw input variables based on spatiotemporal information from each valid RO sample. A uniform sampling strategy was utilized to divide the entire dataset with the aim of ensuring the inclusion of ionospheric information from different solar activities in the training procedure. In summary, the collected RO measurements were partitioned into three sets for the purposes of training, validation, and testing, i.e., the sampled data, about 20%, were designated as the test set (937592 samples), while the remaining data were utilized as the





training (3562999 samples) and validation sets (187527 samples). Since the S4max index correlates well with foEs measured from global ground-based ionosonde (Lei et al., 2007; Yu et al., 2019, 2020), We selected ionosonde data from Beijing, China to validate the generalizability of the model. More detailed information about the raw input variables and preprocessing are described in the S1.3 and Table S1.

## 2.2 Algorithms

The random forest (RF) algorithm is a non-parametric statistical learning method that uses an ensemble of decision trees and can be applied to both regression and classification problems (Breiman, 2001; Svetnik et al., 2003). The principle of RF is done with respect to classification and regression trees (CART) model strategy (Breiman, 2017). To construct an individual decision tree, smaller sub-groups of observations of the predictors, also known as features, are created using optimal decision rules, or split rules. It balances the prevalent overfitting problem, minimizes variance, and ensures improved accuracy by creating

several trees for different subsets of the data points. These groups are split in a way that maximizes the difference in predicted variables between the groups while also maximizing their homogeneity within the groups. Creating several trees for different subsets of the data points balances the prevalent overfitting problem, minimizes variance, and ensures improved accuracy. In regression problems, the predicted value of the RF is determined by the average aggregation of the all decision trees.

The RF ranks the input variables by a variable importance measure, which reflects the impact of the input variables on the

output based on the prediction accuracy (Breiman, 2017). We utilized this approach to perform a preliminary importance analysis of all input variables, facilitating the subsequent selection of effective input information for training. The samples for constructing the RF model were obtained through a process of downsampling from the original training set. The predictive performance of the RF model is improved by increasing the tree strength and decreasing the number of correlations among trees. Empirical studies have previously shown that the maximum RF model performance is often achieved by the first 100

trees, i.e., a larger number of trees in a forest can not necessarily improve performance (Oshiro et al., 2012; Couronné et al., 2018). Therefore, a base model of 100 trees was selected for regression task. A sample size of at least two data points to split an internal node and at least one point is needed for a terminal node.

In the recent years, deep neural network have seen preliminary success in climate science, meteorology, and hydrology, resulting in improved predictive skills and the development of methods to investigate the spatiotemporal dependencies (Ham et al.,

2019; Liu et al., 2023). Sufficient evidence reveals that network depth is of crucial importance (Simonyan and Zisserman, 2014; Szegedy et al., 2015). Excessive depth in neural networks can lead to the notorious problem of vanishing or exploding gradient (Bengio et al., 1994; Glorot and Bengio, 2010). This issue arises when the gradients of the loss function with respect to the model parameters become too small or too large, during backpropagation. As a result, the learning process can slow down significantly or even fail to converge. The Residual network (ResNet) (He et al., 2016) is a type of neural network that

alleviates this problem of training deep learning networks by using skip-connections in every stacked block, which provides alternative paths for original and derived features, rendering training faster and more efficient. We proposed the SELF-ANN model, which has incorporated dense blocks at the front and back ends to adapt to input data, based on the residual network architecture, as shown in Fig. 1. The detailed structure of the SELF-ANN module can be found in Table S2.





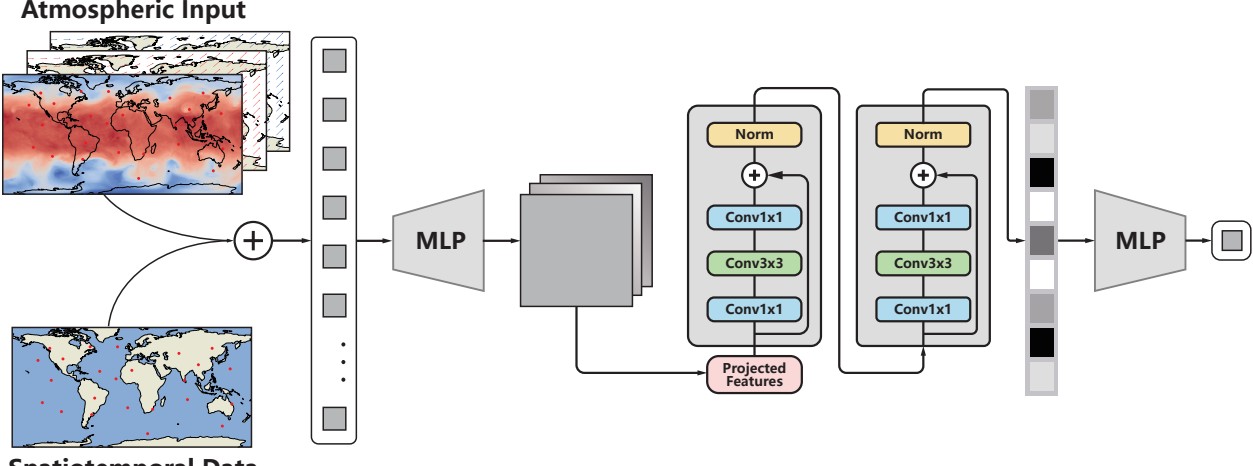

**Figure 1.** Overall framework of SELF-ANN (Sporadic E Layer Forecast using Artificial Neural Networks). MLP represents several dense layers composed of multilayer perceptrons. The conv1×1 and conv3×3 denote convolution layers with convolutional kernels of 1 and 3, respectively. Norm refers to batch normalization operation. The number of neurons depicted is not representative.

The lower atmospheric data and spatiotemporal information are first subjected to feature fusion, followed by standardization, a
regular preprocessing operation, before being fed into the model. Figure 1 illustrates the SELF-ANN architecture, which con-
sists of the convolutional layer (conv1×1 or conv3×3), batch normalization (norm) units (Ioffe and Szegedy, 2015), residual
blocks (ResBlock), fully connection layer composed of multilayer perceptrons (MLP). Each ResBlock consists of 3 convo-
lution layers, followed by a rectified linear activation units (ReLU) activation function (Nair and Hinton, 2010) and batch
normalization, where it protects the integrity of geoscience information through skip connections. Such skip connections, or
called shortcuts, are those skipping one or more layers and simply perform identity mapping, as expressed by the following
equation:

$$\mathbf{y} = \mathcal{F}(\mathbf{x}, \{W_i\}) + W_s\mathbf{x} \tag{1}$$

Here $\mathbf{x}$ and $\mathbf{y}$ are input and output vectors of the layers considered. The function $\mathcal{F}(\mathbf{x}, \{W_i\})$ represents the learned abstract
mapping. The $W_s$ is a linear projection to match the dimensions of $\mathbf{x}$ and $\mathcal{F}$. The aggregate features via shortcuts will perform
batch normalization, which is an important way to address the problem of vanishing/exploding gradients. In the context of batch
normalization, the calculation of the mean and variance of a batch of images can be expressed using Eq. 2 and 3, respectively.
The resulting values are subsequently utilized in Equation 4 to perform normalization, followed by Eq. 5 for scaling and





shifting. The detailed formulas are listed as follows.

$$\mu_n = \frac{1}{n}\sum_{i=1}^{n} x_i \tag{2}$$

$$\sigma_n^2 = \frac{1}{n}\sum_{i=1}^{n}(x_i - \mu_n)^2 \tag{3}$$

$$\widehat{x}_i = \frac{x_i - \mu_n}{\sqrt{\sigma_n^2 + \varepsilon}} \tag{4}$$

$$\hat{y}_i = \gamma\widehat{x}_i + \beta \tag{5}$$

where $n$ represents the batch size, $x_i$ refers the $i$th value of $x$, $\mu_n$ and $\sigma_n^2$ denote the mean and variance of $x$, $\widehat{x}_i$ means the output of batch normalization, $\varepsilon$ represents an infinitely small value, $\gamma$ and $\beta$ represent the parameters of scaling, $\hat{y}_i$ is the final scaled output. The ReLU activation function is used after batch normalization. As expressed in Eq. 6, the operation outputs 0 when $\hat{y}_i \leq 0$, and conversely outputs a linear mapping when $\hat{y}_i > 0$.

$$\mathrm{ReLU}(\hat{y}_i) = \begin{cases} \hat{y}_i & \text{if } \hat{y}_i > 0 \\ 0 & \text{if } \hat{y}_i \leq 0 \end{cases} \tag{6}$$

### 2.3 Evaluation Metrics

The present study employs multiple evaluation metrics to assess the performance of SELF-ANN. These metrics are calculated by quantifying the discrepancies between the observations and the model output. The following metrics are utilized:

$$\begin{cases} \mathrm{SCC} = \dfrac{cov\left(R(\hat{y}_i), R(y_i)\right)}{\sigma_{R(\hat{y}_i)} \cdot \sigma_{R(y_i)}} \\[2ex] \mathrm{ME} = \dfrac{1}{N}\sum_{i=1}^{N}(\hat{y}_i - y_i) \\[2ex] \mathrm{MAE} = \dfrac{1}{N}\sum_{i=1}^{N}|\hat{y}_i - y_i| \\[2ex] \mathrm{RMSE} = \sqrt{\dfrac{1}{N}\sum_{i=1}^{N}(\hat{y}_i - y_i)^2} \end{cases} \tag{7}$$

where SCC is the spearman correlation coefficient; ME is the mean error; MAE is the mean absolute error; and RMSE is the root mean square error. $N$ is the number of the RO samples in the test data. $y_i$ and $\hat{y}_i$ are the target values, i.e., the S4max index, of the observations and predictions, respectively. $R(y_i)$ and $R(\hat{y}_i)$ are the rank values of those two variables. $cov(\cdot)$ means the covariance operation. $\sigma_{R(y_i)}$ and $\sigma_{R(\hat{y}_i)}$ are the standard deviations of $R(y_i)$ and $R(\hat{y}_i)$, respectively. Moreover, the Altman-Bland method was employed to evaluate the discrepancies between the measurements and the predictions (Altman and Bland, 1983). The method was also used to calculate the 95% confidence limits of agreements for the estimation (average difference $\pm$ 1.96 standard deviation of the difference). The detailed calculation steps of the Altman-Bland method can be found in S1.2.





## 3  Results

The SELF-ANN source code was implemented in Python with PyTorch package and trained on a remote server node with
eight 80 GB NVIDIA A100-SXM4 graphics processing units (GPUs), dual 32-core Intel® Xeon® Platinum 8358 CPUs @
2.6 GHz, and 1 TB of RAM. The mean absolute error loss function is used during the training process. Batch normalization
is implemented to accelerate training by reducing internal covariate shift (Ioffe and Szegedy, 2015). The Nadam optimizer is
employed to optimize the predefined loss function where the approach is robust in deep learning (Ruder, 2016). The learning
rate is initially set at 0.0002, a critical parameter in this research, and the batch size is set to 256. In addition, We used a
technique for tweaking the learning rate scheduler, called ReduceLROnPlateau, to vary the learning rate depending on the
number of epochs to accelerate the model's convergence (Bowling and Veloso, 2002). The technique monitors a quantity and
reduces the learning rate when that quantity stops improving.

### 3.1  The Importance of Input Variables

The random forest regression algorithm was utilized to evaluate the input variables, with their importance rankings established
through an analysis of their impurity (Breiman, 2001; Svetnik et al., 2003). During a downsampling operation for all data,
we take the 1562941 vectors as input $x$, where each vector comprises $n$ parameters of the lower atmosphere information and
external forcing (solar and geomagnetic), and the corresponding ionospheric scintillation index served as the output $y$. As it
is currently unfeasible to determine the neutral wind fields in the E region, we have instead considered the lower atmospheric
variables as potential seed sources that may influence the wind fields in the ionospheric region (Kazimirovsky et al., 2003; Yiğit
et al., 2016). The lower atmospheric parameters, including variables such as wind, temperature and geopotential, were collected
from the ERA5 reanalysis dataset. Together with the spatiotemporal information of the source-region of samples and external
forcing such as geomagnetic and solar activities, the dimension of $x$ space is 52 ($n = 52$). All of the aforementioned features
may play a vital role in predicting the ionospheric irregularities, however, explicitly removing irrelevant features improves both
dimensionality reduction and noise elimination. The procedure of training random forest model and all input variables used for
importance ranking can be found in S1.1 and Table S1, respectively.

Figure 2 illustrates the importance ranking results of different input variables. The high mean decrease impurity (MDI) values
obtained from the spatiotemporal location information of ionospheric irregularities, as depicted by local time, altitude, lati-
tude, and longitude in the figure, are not unexpected, given a strong correlation with the intensity of the Es. These findings
are consistent with previous research on the morphology of Es, such as the diurnal variation of the scintillation occurrence
(Ogawa et al., 1989; Yu et al., 2021b), and the global scale distribution of Es intensity (Yu et al., 2019). The zonal wind in
the stratosphere ranks fifth in importance for the investigated parameters, following the spatiotemporal information. It could
act as a seed source, affecting dynamical processes in the E-region (Goncharenko et al., 2010). Additionally, geopotential and
temperature variables also exhibit a significant contribution, which is an interesting result. This phenomenon can be attributed
to the impact of upward propagation of internal atmospheric waves (planetary waves, tides and gravity waves), which act as
an essential source of energy and momentum on the ionosphere (Yiğit et al., 2016; Kazimirovsky et al., 2003). Solar activity





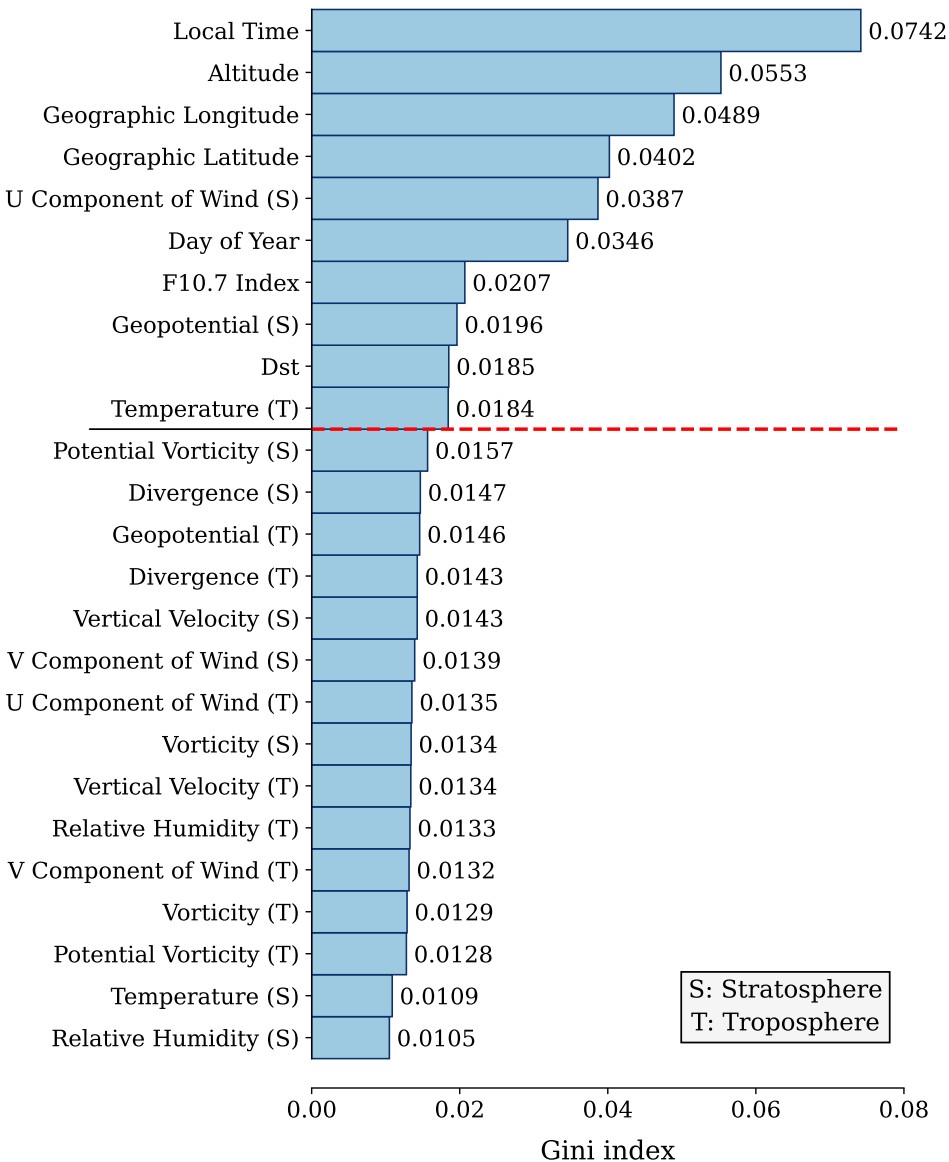

**Figure 2.** The importance ranking of input variables is based on the mean decrease impurity, which is computed by the random forest regression algorithm. The letters S and T represent the stratosphere and troposphere, respectively. The part above the red dashed line represents the variables that were selected for subsequent training.





and geomagnetic disturbances also play significant roles with relatively high MDI scores, possibly attributable to the influence of external solar and geomagnetic forcing from above on the terrestrial system (Pedatella, 2016; Yiğit et al., 2016). In this study, we selected the variables above the red dotted line, as shown in Fig. 2, as the input variables for subsequent SELF-ANN training.

## 3.2 Results of Reconstructing E-region Morphology

The results of the global distribution of the test set are presented and discussed in this subsection. In the test data, we analysed a total 937592 COSMIC radio occultation profiles. The plots presented in Fig. 3 show the global morphology of Es intensity from observation (Fig. 3a) and SELF-ANN (Fig. 3b). The data is binned in a $2.5° \times 2.5°$ geographic latitude/longitude grid. To evaluate the regions with significantly higher Es intensity on the globe, we calculated the residual value (mean $\pm$ std) of observation and the proposed model, depicted by the red shaded area. Inspecting Fig. 3 shows that the predictive results of the SELF-ANN exhibit consistency with observations on a large global scale, particularly in capturing the regional distribution feature with higher Es intensity. In general, the distribution of Es demonstrates a strong dependence on the geomagnetic latitude (Yu et al., 2019, 2020, 2021a). For example, the strong magnetisation of the electrons near the dip equator results in the missing of Es intensity. The wind shear mechanism which is responsible for sporadic E formation does not exist close to the geomagnetic equator. The SELF-ANN provides astonishingly similar predictive results regarding the missing of Es intensity near the geomagnetic equator.

Figure 4 presents global distributions of the seasonal mean intensity of the Es layer for different seasons in the test data. The presented figure demonstrates that the variability of the Es intensity is primarily influenced by seasonal variations, with its maximum in the summer hemisphere and minimum in the winter hemisphere. The morphology of the Es layers in the four seasons from the proposed model agree with the observations. The right panels depicting the mean values along the geographic latitude reveal that the Es layer with S4max values greater than 0.5 is mainly distributed at midlatitudes, as illustrated in Fig. 4g. The Es layer exhibits a weakened presence in the lower latitudes in both hemispheres, and its minimum strength, with S4max values less than 0.2, is located at $60°N$ latitude in winter. Moreover, the intensity of Es layer in the northern summer hemisphere is much higher than it is in the southern summer hemisphere. This is probably associated with the lower thermospheric meridional circulation which flows from the summer to the winter hemisphere (Yu et al., 2021b). These shaded blocks with colors indicate the regions with extreme maximum or minimum values for comparison. For example, the deep blue block in the Fig. 4d covers the region near the geomagnetic equator, which exhibits relatively weak Es intensity. The absence of Es intensity along the geomagnetic equator, where the proposed model successfully captures this feature, emphasizes the significant role of geomagnetic control in the formation of the Es layer. As depicted in Fig. 4g and 4h, the results of the SELF-ANN model display certain disparities when compared to the observations particularly at high latitudes. A possible explanation for this inconsistency could be attributed to the limited availability of training data in these regions (Tian et al., 2022; Yu et al., 2022). Overall, the climatological features of Es layers are reproduced by the SELF-ANN, indicating its capability to accurately capture the morphology and overall evolution of the ionospheric Es.

Furthermore, a statistical analysis of altitude profiles is presented. Figure 5 shows that the altitude-local time, altitude-day of

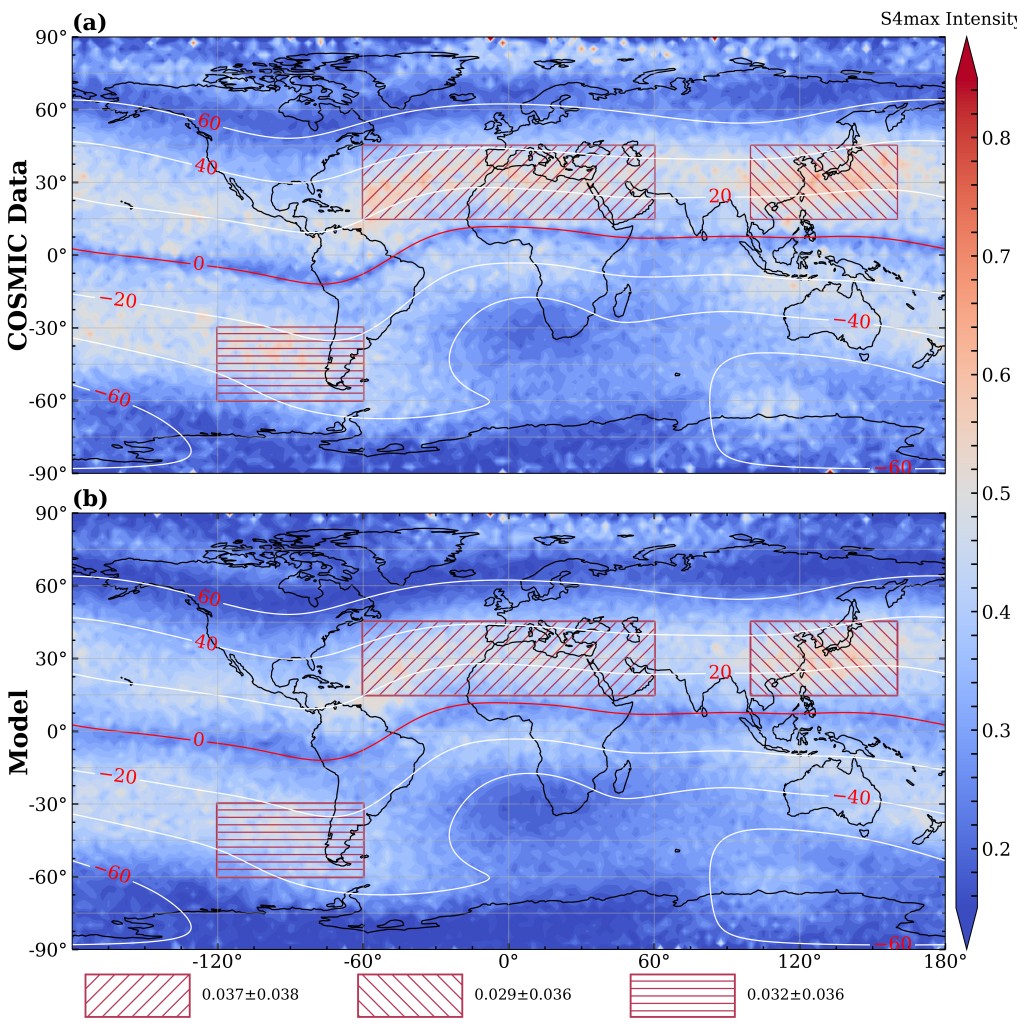

**Figure 3.** Global distribution of sporadic E intensity from test data. (a) COSMIC RO measurements. (b) SELF-ANN prediction. The red shaded areas represent the mean $\pm$ standard deviation of the residuals of the model and observations. Thin white curves signify geomagnetic latitude contours and the thick red curve is the geomagnetic equator. The geomagnetic latitude was calculated from the International Geomagnetic Reference Field (IGRF). They have a latitude/longitude resolution of $2.5° \times 2.5°$.




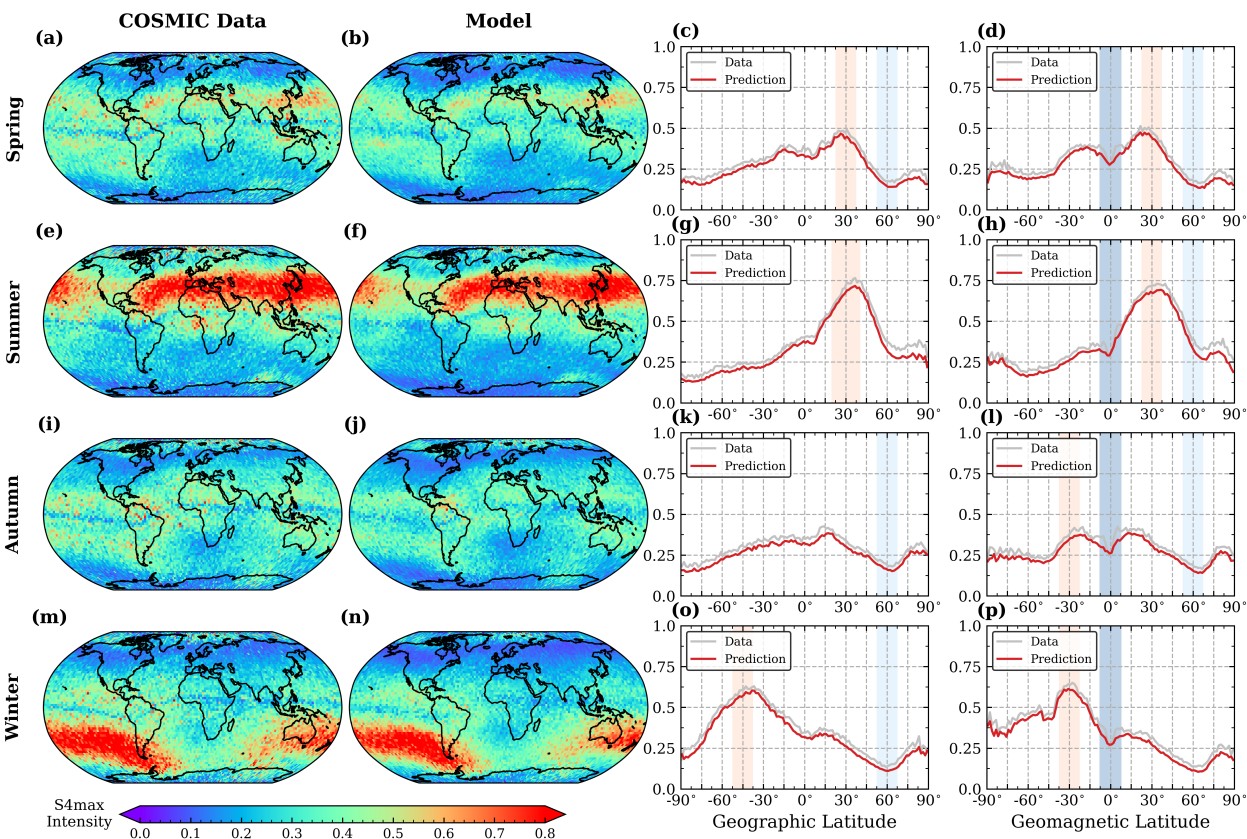

**Figure 4.** Global distributions of the intensity of the Es layer for different seasons, represented by S4max from COSMIC observations and the SELF-ANN outputs in the period of test data. The different rows in the figure represent spring (March, April, and May), summer (June, July, and August), autumn (September, October, and November), and winter (December, January, and February), respectively. The third and fourth panels represent the mean values along geographic latitude and geomagnetic latitude, respectively. The shaded blocks with colors indicate the regions with extreme maximum or minimum values for the purpose of comparison between COSMIC observations and model predictions.



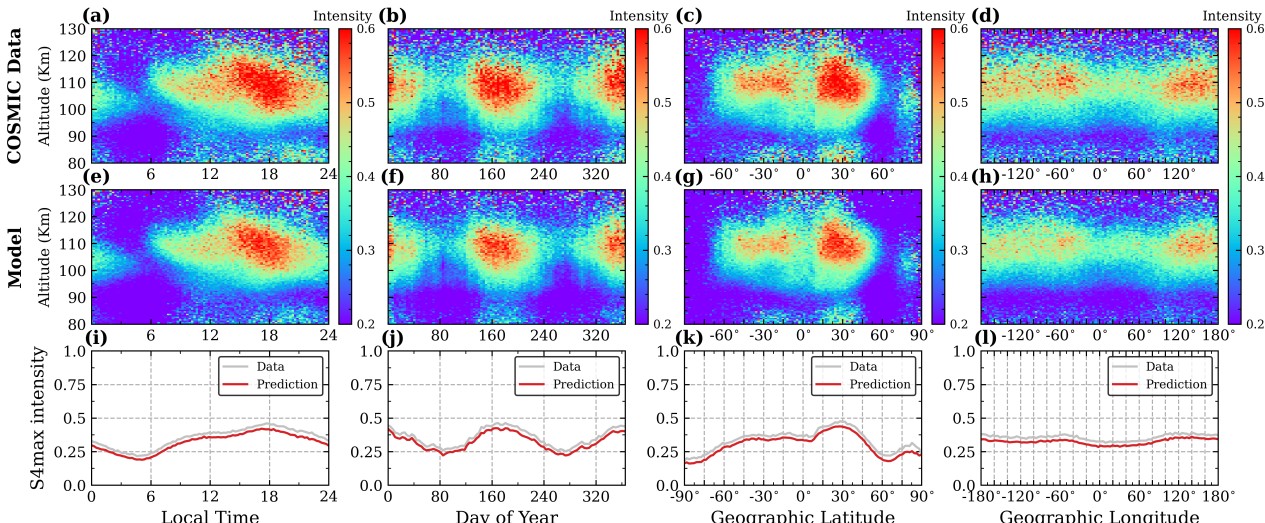

**Figure 5.** The altitude-local time, altitude-day of year, altitude-latitude, and altitude-longitude distributions of S4max from COSMIC RO measurements in the period of test data. (a)-(d) and (e)-(h) correspond to the observed data and the predictions made by SELF-ANN, respectively. (i)-(l) represent the average intensity of S4max.

255  year, altitude-geographic latitude, and altitude-geographic longitude distributions of Es intensity from COSMIC observations and SELF-ANN in the test data. In general, the high S4max values exhibit a predominant spatial distribution within the 100-120 km altitude range, as illustrated in Fig. 5a-5d. It is noteworthy that the SELF-ANN output has effectively reproduced this distribution feature at the corresponding altitudes. Figure 5a reveals that the altitude distribution of Es varies with local time (LT). Specifically, the intensity of Es is observed to be weaker at approximately 6 LT, while it is stronger at approximately

260  18 LT. The SELF-ANN demonstrates the ability to successfully capture this diurnal and semidiurnal variation trend in the Es distribution. In addition, the proposed model has reconstructed the annual and semiannual variations found in the seasonal-to-interannual time series of S4max. For instance, in the Northern Hemisphere, the intensity of Es is higher during the summer season compared to other seasons, as depicted in Fig. 5f. As shown in Fig. 5c, the distribution of Es shows an asymmetry between the northern and southern hemispheres, with the maximum value of S4max observed at midlatitudes. Moreover, the

265  distribution map of altitude-geographic longitude displays a pattern of wave-like structures (Fig. 5d) (Liu et al., 2021; Yu et al., 2022). These features of the morphology of Es layers are accurately reconstructed by the SELF-ANN.

Prior studies have established that the occurrence rate of the Es layer is a commonly utilized metric for statistical analysis of scintillation events (Arras et al., 2008; Yu et al., 2019; Ye et al., 2021). In light of this, we conducted an analysis of amplitude scintillation statistics using data collected by COSMIC satellites and predictions made by SELF-ANN among the test data.

270  To analyze the Es layer scintillation characteristics, the radio occulation event is considered to include a disturbance, the occurrence of a sporadic E layer, in case the respective S4max intensity exceeds a defined threshold of 0.2 (Arras et al., 2008; Arras and Wickert, 2018; Tian et al., 2022). Figure 6 displays the distribution between altitude, local time, and geomagnetic





latitude in relation to the intensity of amplitude scintillation and the occurrence rate of the Es layer derived from observations and predictions, accompanied by the corresponding histogram. Within the figure, the blue and gray data points showcase the respective observed and predicted values, whereas the corresponding blue and gray curves highlight the calculated occurrence rate of Es, respectively. In general, the COSMIC observations and SELF-ANN predictions are in good agreement. The histograms in Fig. 6a-6c present the percentage of RO events with respect to altitude, local time, and geomagnetic latitude, as derived from COSMIC data after binning. The majority of S4max index values for scintillation are below 0.5 (Fig. 6d), with the highest occurrence rate of Es observed at an altitude of approximately 110 km (Fig. 6e). At 6 LT, the occurrence rate of Es is found to be comparatively low, whereas at 18 LT, it shows an increase (Fig. 6f), which is consistent with the results in Fig. 5i. This can be attributed to the neutral winds controlled by the solar tides in the E region that may dominate and thus govern the diurnal and sub-diurnal variability and descent of the layers through their vertical wind shears (Kazimirovsky et al., 2003; Yiğit et al., 2016). The wind shear theory indicates that the close connection between the formation of Es layer and geomagnetic latitude for the midlatitudes (Fig. 6g). The occurrence of Es is comparatively inhibited at the geomagnetic equator due to the ions fail to converge vertically into a layer when the geomagnetic field is horizontal. As the geomagnetic inclination rises, the convergence of ions in the vertical direction gradually amplifies, leading to a rise in the occurrence rate of Es. In regions of high geomagnetic latitude, where $\cos I \sim 0$, the vertical velocity of ions is minimal, but the vertical effects of internal waves are effective because of the nature of geomagnetic field being vertical (Haldoupis, 2011; Plane et al., 2015; Yu et al., 2019). It is clear to see that the SELF-ANN is capable of reproducing a similar Es occurrence variation against local time like that seen in the observed data and also captures the difference in Es morphology caused by different physical mechanisms, although its occurrence rate predicted by SELF-ANN is lower than that retrieved from the RO data in high geomagnetic latitude regions in part owing to the paucity of data, as depicted by the scatter and line plots in Fig. 6e-6g.

### 3.3 Quantitative Evaluation and Application

To ensure an accurate assessment of the performance of the SELF-ANN model, we conducted quantitative statistical analyses and compared the results to ground-based observations between 2008 and 2014. The histogram of the discrepancies between the observational data obtained from COSMIC and the model predictions proposed in this study, as depicted in Fig. 7a. The residual distribution of the test set data indicates a satisfactory agreement between the model and observations. Specifically, approximately 77% of the data, depicted by the area in the red box in the figure, show a distribution centered around zero. The evaluation metrics, namely ME, MAE, and RMSE, attained values of -0.03, 0.2, and 0.33, respectively. Figure 7b displays the Altman-Bland plots of test cases from COSMIC observations and SELF-ANN predictions, with a least-squares fit to the scattered points shown as a black line. It is apparent that the data points in the plot of differences versus averages form a horizontal V-shape, with the open end positioned towards the right, suggesting the feasibility of constructing V-shaped limits (Ludbrook, 2010). The UCL and LCL represent the upper and lower bounds of the corresponding 95% confidence limits, respectively. The majority of scattered data points fall within the region bounded by the upper and lower confidence limits, which define the range within 95% differences that can be expected to occur in the scintillation index from the samples (Dewitte et al., 2002;





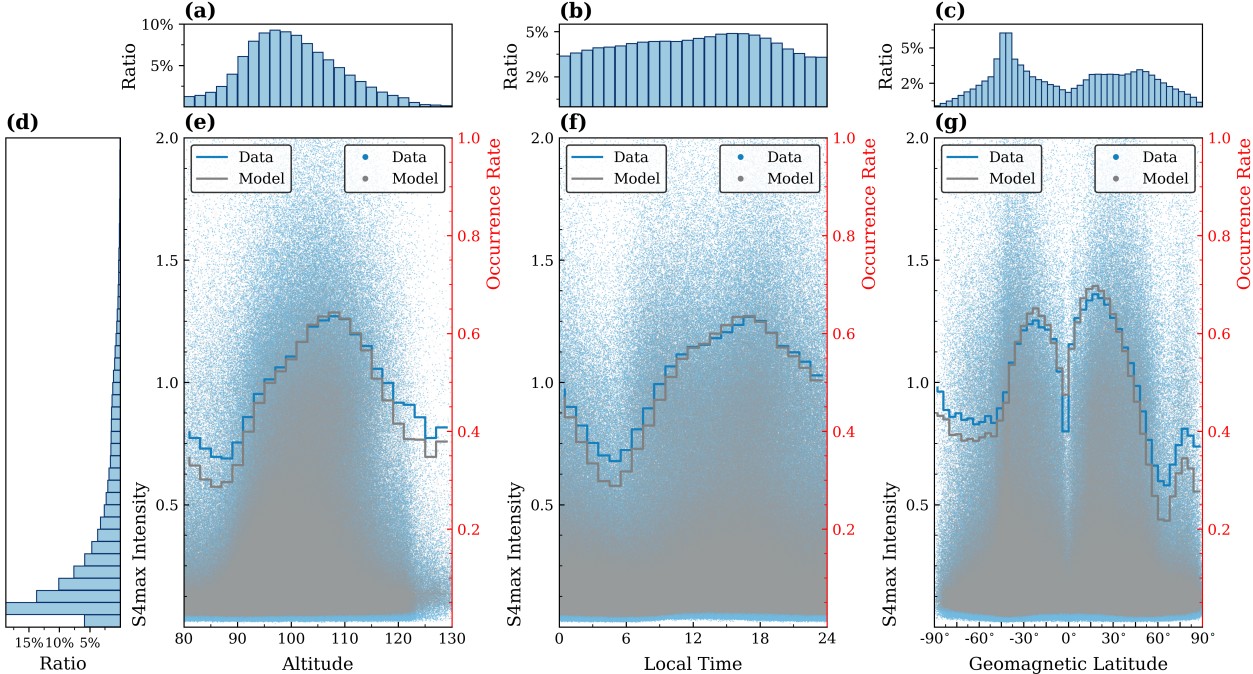

**Figure 6.** Amplitude scintillation statistic measured by the COSMIC satellites (blue) and predicted by SELF-ANN (gray) between 2013 and 2014. The histograms of RO events observed by COSMIC against altitude (a), local time (b), and geomagnetic latitude (c). (d) The histogram distribution of S4max intensity measured by COSMIC at 80-130 km. The S4max intensity versus altitude (e), local time (f), and geomagnetic latitude (g). The blue (gray) curve represents the occurrence rate of S4max $\geq 0.2$ from observations (prediction).

Ludbrook, 2010). This showcases the statistical reliability of the predictive outcomes produced by the presented model. The detailed calculation steps of the Altman-Bland method can be found in S1.2.

The Fig. 7c illustrates a density scatter plot of scintillation index from COSMIC observations and SELF-ANN outputs, with

the corresponding correlation coefficient. The correlation analysis between the model outputs and the observations yields a correlation coefficient of 0.607, indicating a moderate degree of correlation between two parameters. The p value, which is less than 0.05, suggests that the correlation coefficient is statistically significant, given the conventional threshold of 0.05 for p values. The statistical findings reveal that the model has a tendency to overestimate smaller values of scintillation index. This can be attributed to the fact that the model primarily offers a climatological perspective of Es layers, and other electrodynamic

processes that potentially influence the vertical movement of ions, such as neutral wind shear effect, have not yet been accounted for. The results presented in the plot show that the SELF-ANN model is capable of reconstructing Es layers using data from satellite RO measurements, although the prediction of Es layers is severely constrained at present due to a lack of sufficient thermospheric wind data. Furthermore, numerous studies have conducted comparisons between GPS-RO observations of Es layers and ionosonde measurements, revealing that the S4max index exhibits a strong correlation with measurements

obtained from worldwide ground-based ionosondes (Emmons et al., 2022; Hodos et al., 2022; Yu et al., 2020; Gooch et al.,



2020; Yu et al., 2021a). To ensure the potential practicality of the model and to further evaluate its performance, a comparison was made between the S4max predicted from the SELF-ANN model trained on COSMIC satellite RO data and the hourly manually scaled critical frequencies of Es layers, foEs, obtained from a ground-based ionosonde located in Beijing (40.3°N, 116.2°E). Figure 7d shows the scatter plot of the hourly manually scaled foEs from the Beijing ionosonde versus the hourly

S4max scintillation index model outputs in the period 2008-2014, selected within a region of $5° \times 5°$ geographical latitudes and longitudes square. The red line represents ordinary least squares line of best fit, which yields the relation between foEs and SELF-ANN outputs $\text{foEs} = 3.25 + 1.81 \times \text{S4max}$ (correlation coefficient: $r = 0.531$, $p \ll 0.05$). The comparison of the predictions with ground-based observations demonstrates the practical applicability and satisfactory predictive performance of the proposed model. Additional quantitative assessment results with other ground-based stations, namely Mohe (MH453),

Shaoyang (SH427), Sanya (SA418), and Wuhan (WU430) station, can be accessed in Fig. S1-S4 in supplementary material.

In light of its notable predictive performance that was verified through meticulous comparisons with ground-based observations, a SELF-ANN-based application, that is open-source and user-friendly, has been established for the purpose of advancing the forecasting of ionospheric irregularities within the space weather community. Figure 8 shows the graphical user interface (GUI) of the application and its statistical results of predictions. The application initially acquires valid input variables, in-

cluding year, month, day, hour, latitude, longitude, and altitude, as depicted in Fig. 8a. After that, the tool performs forward propagation within simplified SELF-ANN model, thereby generating the desired S4max value. The statistical distribution of the mean S4max intensity, as a function of local time and geomagnetic latitude, is illustrated in Fig. 8b-8c. The variation of mean S4max intensity predicted by the application is consistent with the observational data, as evidenced by the previous Fig. 5. The GUI application is based on a simplified version of SELF-ANN that enables users to generate predictions within sec-

onds of providing necessary inputs (https://github.com/RuleNHao/SELF-ANN). The simplified model ignores external driving factors and focuses simply on spatiotemporal information, offering faster computation speed and a wider valid time scope. To the best of our knowledge, this is the first proposed GUI tool of artificial intelligence for prediction in the sporadic E layer. To summarize, the SELF-ANN-base tool offers several key advantages, including its ability to provide accurate predictions, its independence from a priori assumptions or theories, and its potential for continued improvement as more input events are added

due to the underlying principles of machine learning, making it a valuable contribution to the field of artificial intelligence in the context of space weather prediction.

## 4   Discussion

We show the existence of an implicit connection between ionospheric Es layer and external factors (e.g. lower atmosphere, geomagnetic and solar activity), which can be extracted via the deep convolutional network. The input feature is obtained by

linear interpolation of the spatiotemporal position of the RO event. Each sample consists of local spatial and temporal information, external atmospheric parameters, geomagnetic disturbances, and solar activities, with multi-source data fusion. In this case, we utilized ten input features of important ranking to train a deep learning model that effectively resolves a regression task and reproduces the scintillation index with exact spatiotemporal location. Furthermore, to our knowledge, this is the first



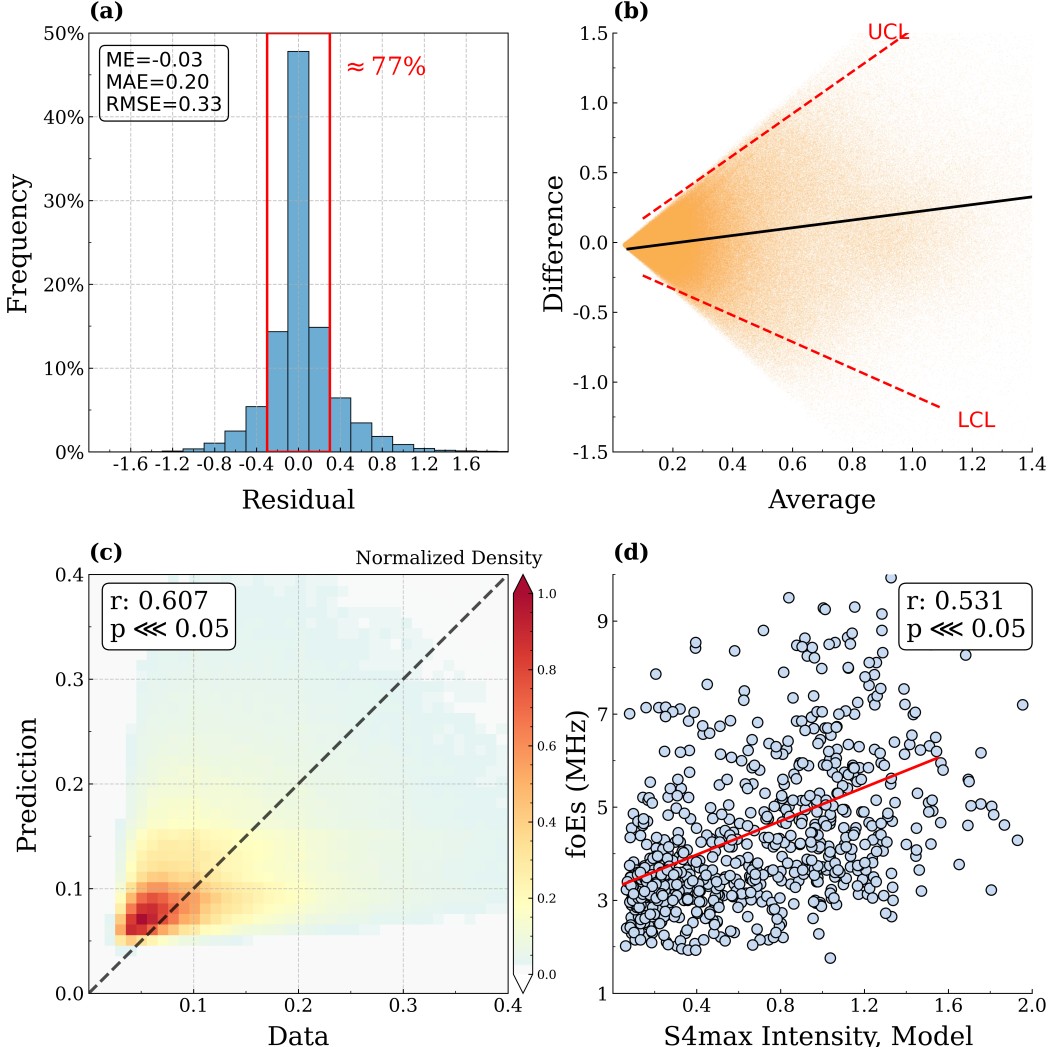

**Figure 7.** Quantitative assessment results of the SELF-ANN and COSMIC observational data, as well as comparative validation of the proposed model with ground-based observations in the test data. (a) Histogram of residuals of the amplitude scintillation index between the observation and prediction. The metrics, including mean error (ME), mean absolute error (MAE), and root mean square error (RMSE), are respectively marked in the upper left part. (b) The Altman-Bland plot of the SELF-ANN prediction in the test data. The UCL and LCL correspond to upper and lower 95% confidence limits for the Altman-Bland limits of agreement. The black line, ordinary least square line of best fit. (c) Density scatter plot of the scintillation index from COSMIC observations and the model outputs. (d) Scatter plot of the hourly manually scaled foEs measured by an ionosonde at Beijing versus the hourly scintillation index from SELF-ANN outputs in the period 2008–2014. The red line represents ordinary least square line of best fit.





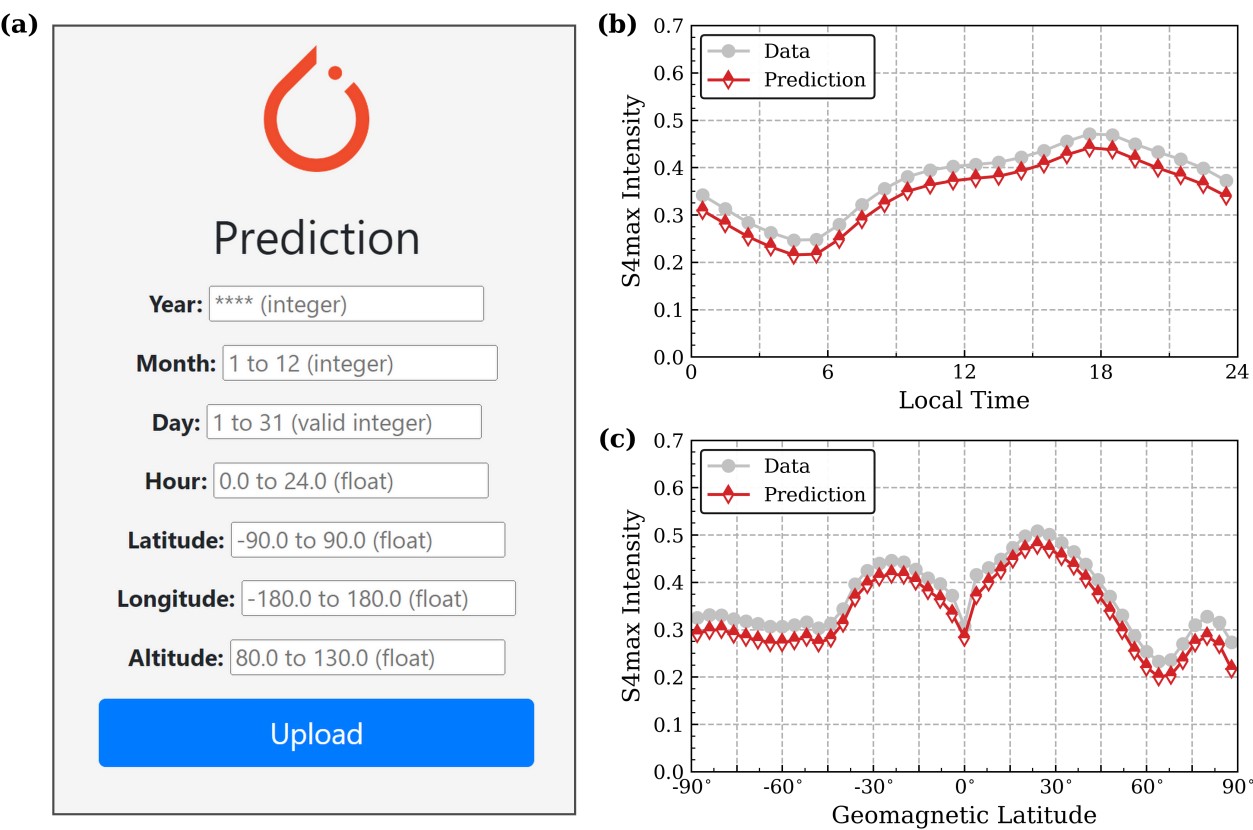

**Figure 8.** User interface of the SELF-ANN application and its output results. (a) Graphical user interface of the application based on the proposed model. The statistical distribution of mean S4max intensity against local time (b) and geomagnetic latitude (c), comparing SELF-ANN tool outputs and observations.

deep learning-based GUI application developed for the Es layer reconstruction, which represents a significant contribution to

the community in terms of forecasting ionospheric irregularities.

Taking into consideration factors such as data transmission, computational speed, and data time scale, the community-accessible tool is based on a simplified version of SELF-ANN, which provides the wider valid time scope spanning approximately one full solar cycle (https://github.com/RuleNHao/SELF-ANN). The backend inference of the simplified model requires only temporal and spatial input parameters, obviating the requirement for other external driving factors (e.g. ERA5 reanalysis dataset), for

the purpose of providing long-term forecast results covering solar activity cycle with fast computational capabilities. The tool is equipped with a GUI (Fig. 8), which affords users the ability to customize valid time and space parameters and view the corresponding output. The main advantage of using SELF-ANN is its capacity to provide the S4max scintillation index in an intuitive way through simple parameter configurations without technical barriers. In addition, the consistency of the model prediction and the independent observations from several ground-based ionosondes ensures that the proposed tool is sufficiently





generalized and reliable.

Our model architecture was motivated by ResNet networks used for regression tasks. A total of 4691412 available data samples were extracted from COSMIC occultation events, each accompanied by corresponding variables representing external driving factors. The input variables undergo a conversion process, wherein they are transformed into the one-dimensional input array. Since an increase of data points in the input space (e.g. using the all available lower atmosphere data) did increase the compu-

tational needs above the available resources, we restricted to the maximal depth of first input layer. Therefore, we utilized the random forest regression algorithm, a conventional machine learning technique, to perform initial screening of the data. This approach not only facilitates the identification of more efficient input information for model training, but also provides a degree of insight into the interpretability of the data. No further attempts were made to compare different architectures. The main goal of our work was to demonstrate that the implicit mapping relationships contained between external factors (e.g., lower wind

data, temperature, solar activity, etc.) and the ionospheric Es layer can be extracted by deep learning networks. Future research should focus on the further optimization of this approach, for instance, by integrating the wind data in the E region that is crucial to the Es formation, modifying filter settings, and adjusting network architecture. In addition, the exploration of model interpretability is an important issue for future work.

This study builds upon the research conducted by (Tian et al., 2022) and aims to extend their findings. Compared to (Tian

et al., 2022) findings, our results have demonstrated significant advancements. By treating each radio occultation event as a sample, we have eliminated the error caused by interpolation and improved the accuracy of the model. Furthermore, we have briefly explored the importance of input information in explaining the predictive capability of the model, utilizing the random forest approach. This finding provides valuable insight into the interpretation of the model, facilitating a more comprehensive understanding of its predictive power. The comparison with ground-based ionosonde observations provides strong evidence

for the robustness of the model and its potential for practical applications. In Fig. 4 and 6, the analysis reveals that the Es layers at high altitudes and latitudes exhibit the significant disagreements between the model outputs and the observational data, notwithstanding the model's ability to reconstruct the morphology of the Es layers. A limitation of the model is its primary focus on describing the meridional migration of Es layers within the midlatitudinal range, resulting in a deficient representation of the high altitude and latitude Es layers. Therefore, we will address the shortcomings of the model in the future work by

including more input factors (e.g., strong wind shears, fast solar wind streams, equatorial electrojet current plasma, atmosphere perturbations in E-region, and variations in meteor flux).

In sum, an implicit relationship exists between external driving factors and ionospheric irregularities, and deep convolutional networks can extract latent features from actual observations beyond traditional methods. Deep nets might also substitute or complement human guided feature extraction and knowledge discovery in other specialties where spatiotemporal data are

ubiquitous, including remote sensing and geographic information science, geodesy, atmospheric science, hydrological earth science, and planetary science. This approach can also be applied to distinguish extreme weather events and is expected to contribute to the prediction of space weather.



## 5   Conclusions

In this research, we have introduced a new tool, referred to as SELF-ANN (Sporadic E Layer Forecast using Artificial Neural
Networks), which has been purposefully designed to forecast the ionospheric Es layer by implementing a multi-source data
fusion approach based on deep learning techniques. The model architecture of our proposed approach is derived from ResNet,
with several modifications made to align with the dataset for both accuracy and robustness. The training dataset used in this
study consists of ionospheric Es layer data from COSMIC RO measurements, atmospheric perturbation information from
ERA5 data, the collected geomagnetic disturbances and solar activities, among others. We performed an initial exploratory
analysis of the interpretability for all the input data using conventional machine learning method, with a specific focus on
random forest regression. The findings suggest that local time, altitude, and geographic coordinates play a more significant
role in the reconstruction of the Es layer, in comparison to other parameters. Following a ranking of feature importance, we
identified and selected several highly important features, including the spatiotemporal information, zonal wind, temperature,
geopotential, the geomagnetic disturbances and solar activity, from all the candidate input information for use in model training.
The well-trained model has the capability to effectively capture the implicit relationships between external driving factors, such
as local atmospheric variations, and the ionospheric irregularities. Moreover, the high-level features of large-scale seasonal
variations in Es layers are also efficiently learned, verifying the feasibility, effectiveness, and generalization capability of the
proposed framework.

Multiple quantitative evaluation metrics were employed to evaluate the performance of the model. The evaluation resulted
in achieving a MAE value of 0.2, RMSE of 0.33, and correlation coefficient of 0.607, indicating that the model exhibits
superior predictive performance. In order to assess the generalization ability of the proposed model, ionosonde data collected
from Beijing, China were utilized. The analysis demonstrated that the correlation coefficient between the hourly foEs and
model outputs is 0.531, which ensures the feasibility of practical applications. Therefore, we develop for the first time a web-
based open-source ionospheric E-region forecast application that is user-friendly and can potentially aid researchers in their
investigation of ionospheric irregularities. The proposed tool is expected to make a significant contribution not only to the
prediction of extreme space weather events but also to open up new possibilities for the application of artificial intelligence in
upper atmospheric and ionospheric physics.

*Code and data availability.*   The reanalysis data used for part of the input in this study are available from ECMWF Reanalysis v5 (ERA5)
(https://www.ecmwf.int/en/forecasts/datasets/reanalysis-datasets/era5). The COSMIC RO data were downloaded from the FORMOSAT-
3/Constellation Observing System for Meteorology Ionosphere and Climate (COSMIC-1 (https://www.cosmic.ucar.edu/). Geomagnetic and
solar activity data are obtained from the OMNI Goddard's Space Physics Data Facility (https://omniweb.gsfc.nasa.gov/). The ionosonde
data are available from the Data Centre for Meridian Space Weather Monitoring Project (https://data.meridianproject.ac.cn/data-directory/)
and the National Space Science Data Center, National Science & Technology Infrastructure of China (http://www.nssdc.ac.cn). The GUI
application and source code used in this work can be found in GitHub (https://github.com/RuleNHao/SELF-ANN).



*Author contributions.* PT designed the study and wrote the manuscript. BY provided early ideas and the manually scaled ionospheric observation. BY and PT contributed to the comments on an early version of the manuscript. XX and JW contributed to the discussion of the results and the preparation of the manuscript. HY performed the part of analysis results. HY and TC provided an advisory role with analyzing the data. All authors discussed the results and commented on the manuscript at all stages.

*Competing interests.* The authors declare that they have no conflict of interest.

*Acknowledgements.* We acknowledge the Constellation Observing System for Meteorology, Ionosphere, and Climate (COSMIC) Data Analysis and Archive Center (CDAAC) for providing COSMIC radio occultation data. Moreover, we particularly thank the European Centre for Medium-Range Weather Forecasts for providing the ERA5 reanalysis data. We acknowledge use of NASA/GSFC's Space Physics Data Facility's OMNIWeb (or CDAWeb or ftp) service, and OMNI data. The authors acknowledge the Chinese Meridian Project, the Solar-Terrestrial Environment Research Network, the Geophysics Center, National Earth System Science Data Center at BNOSE, IGGCAS, and
the National Space Science Data Center, National Science and Technology Infrastructure of China for providing the ionosonde data. The training of the model in this manuscript were done on the supercomputing system in the Supercomputing Center of University of Science and Technology of China. This work is supported by the National Natural Science Foundation of China (42125402, 42188101, 42074181), the Project of Stable Support for Youth Team in Basic Research Field, CAS (YSBR-018), the National Key R&D Program of China (Grant No. 2022YFF0503703), the Innovation Program for Quantum Science and Technology (2021ZD0300300), the Frontier Scientific Research
Program of Deep Space Exploration Laboratory (Grant No. 2022-QYKYJH-ZYTS-016), and the Joint Open Fund of Mengcheng National Geophysical Observatory (Grant No. MENGO-202207). J.W. was funded by the Fundamental Research Funds for the Central Universities.



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
