# Peer review of "Ionospheric Irregularities Reconstruction Using Multi-Source Data Fusion via Deep Learning"

_EGUsphere, 2023_

## Author Comment (AC1)

**Ionospheric Irregularities Reconstruction Using Multi-Source Data Fusion via Deep Learning – Tian et al.**

→ response to the editor and reviewer

**NB:** In the followings, the comments from the reviewers are written in **black** and our responses in **green**.
* * *
**Reviewer #1 (Formal Review for Authors):**

This is a well written article that provides important insight into the complicated drivers of sporadic-E. The developed model is the best to date at capturing the spatial and temporal variations, thereby providing a significant contribution to the field.

→ We extend our gratitude to the reviewer for the assessment and acknowledge this positive evaluation.

Key Comments
(1) In the beginning of Section 3.2 Results of Reconstructing E-region Morphology, it would be helpful to provide more information on the COSMIC data used to create the model. For example, what CDAAC file-types are you using and what is the time range of data used for analysis?

→ We appreciate the constructive suggestion and have provided a more detailed information about COSMIC data (**Lines 252-259**).

→ We used COSMIC ionospheric S4 scintillation index and auxiliary data (level 1b datasets) from 2008 to 2014 in this study. Each file contains continuous S4 data (one value each second based on 50 Hz internal receiver sampling) from one GPS satellite. The data format called scnLv1 format can be found on the website ([https://cdaac-www.cosmic.ucar.edu/cdaac/cgi_bin/fileFormats.cgi?type=scnLv1](https://cdaac-www.cosmic.ucar.edu/cdaac/cgi_bin/fileFormats.cgi?type=scnLv1)). The file provides s4max (the maximum S4 index based 1-sec), alttp_s4max (altitude of tangent point position at maximum S4 index), lattp_s4max (latitude of tangent point position at maximum S4 index), lontp_s4max (longitude of tangent point position at maximum S4 index), and lcttp_s4max (local time of tangent point position at maximum S4 index). We removed invalid data and collected S4max data at altitudes between 80-130 km. In the formulation of the proposed model, the filtered data's spatiotemporal attributes, including year, month, day, longitude, latitude, and altitude, were employed as input parameters. Approximately 937,592 samples from the dataset between 2008 and 2014 were utilized for SELF-ANN evaluation and analysis.

(2) Throughout the paper, the data is binned and analyzed over a single variable to show trends. For this binning, do you present the average value of the bin or some other parameter such as the maximum? Also, when the data is displayed as a function of a single variable (such as Figure 5), do you average over all of the other parameters? More information on the binning and averaging would be extremely helpful.

→ Thank you for the suggestion. We have added further clarification to the figure descriptions, particularly for those presenting a single variable (**Lines 261, 271-273, 293-294, 313-315**).

→ For the binning of individual variables, we present the average values of the data, rather than the maximum, whether for observations or model-predicted values.
→ More specifically, the following table lists a portion of the raw data, with corresponding predicted values provided by the model for each COSMIC measurement. As these data are not on a regular grid, we have partitioned them into different bins according to individual variables, in order to present a comparison between the model and observation. When analyzing a single variable's distribution, other parameters are treated as averaged.

| Altitude (km) | Latitude | Longitude | Local Time | S4max (Observation) | S4max (SELF-ANN) |
|---|---|---|---|---|---|
| 96.481003 | 4.990000 | 140.457001 | 9.632000 | 0.495 | 0.499189 |
| 98.782997 | -22.993000 | 155.822998 | 11.032000 | 0.606 | 0.592083 |
| 90.579002 | 10.438000 | -75.638000 | 20.318001 | 0.151 | 0.151524 |
| 104.514999 | -48.216000 | -51.139000 | 21.525000 | 0.538 | 0.522375 |
| 105.690002 | -19.634001 | -66.695000 | 20.577999 | 0.303 | 0.306369 |
| ... | ... | ... | ... | ... | ... |
| 99.264999 | -71.334000 | -42.504002 | 16.403000 | 0.483 | 0.464988 |
| 92.075996 | -29.573000 | 16.091000 | 20.561001 | 0.114 | 0.103630 |
| 98.010002 | -77.949997 | -21.507999 | 18.046000 | 0.055 | 0.066798 |
| 110.815002 | 63.476002 | -171.796005 | 8.944000 | 0.083 | 0.102691 |
| 98.644997 | -47.664001 | -159.522995 | 10.063000 | 1.761 | 1.758963 |

→ For example, in Fig. 5a, data are binned according to specific local time intervals and altitude intervals (each uniformly divided into 100 bins). Each bin corresponds to the average value of all S4max values within that bin. Figure 5e undergoes the same processing, but the data are sourced from the SELF-ANN model. Figure 5i depicts a line plot derived by averaging the grid point data from both Fig. 5a and Fig. 5e along the local time. Moreover, to facilitate visualization, the intervals for altitude, local time, and geomagnetic latitude are respectively set at 2 km, 1 hour, and 4 degrees in the Fig. 6. These clarifications have been incorporated into the revised manuscript.

Minor Comments
(1) Line 78: Please define ECMWF here.

→ Corrected (**Lines 87-91**).

(2) It would be helpful to add commas in the large numbers such as in 937592 in Line 124.

→ Corrected throughout the manuscript (**Lines 135-140, 224-228, 258-259, 422-425**).

(3) Line 126: We => we.

→ Thanks for your careful review. Corrected (**Line 141-142**).

(4) Line 131: It's unclear what you mean by "The principle of RF is done".

→ Corrected (**Lines 146-148**).
→ The purpose of this statement is that the model strategy of RF (Random Forest algorithm) is CART (Classification and Regression Tree), and it will be performed to completion following this strategy. We have made the necessary corrections.

(5) Lines 234-235: I suggest replacing "missing of Es" with "lack of Es".

→ Thanks for your suggestion. Corrected (**Lines 266-270**).

(6) Line 245: Is this supposed to be a winter to summer meridional flow?

→ Thanks for pointing out this typo. The meridional circulation is indeed winter to summer in the lower thermosphere. Corrected (**Lines 280-282**).

(7) Figure 3: It's tough to see the variation from this color scheme because of the few high latitude patches with a large

S4max…perhaps you could change the color bar so the peak S4max color is lower, which would help to show more of the mid-latitude patterns.

→ Thanks for your suggestion. The caption and description of Fig. 3 have been modified (**Fig. 3; Lines 263; Captions Fig. 3**).

→ To enhance contrast, we modified the color scheme and the display range of the color bar. Concurrently, shadows on the boxes were removed and replaced with numerical identifiers, as shown below. Original image on the left, modified image on the right.

---

## Author Comment (AC2)

**Ionospheric Irregularities Reconstruction Using Multi-Source Data Fusion via Deep Learning – Tian et al.**

→ response to the editor and reviewer

**NB:** In the followings, the comments from the reviewers are written in **black** and our responses in **green**.
* * *
**Reviewer #2 (Formal Review for Authors):**

General Comments
This is a high-quality manuscript offering a novel insight into how AI techniques can be used to support the prediction of Es layer morphology. The model is able to capture the Es climatology incredibly well, and therefore represents a significant step forward in the field of Es layer modelling. A number of minor typographical corrections have been suggested to aid readability, but overall, the article is well written and structured.

→ We thank the referee for examining our work and giving a positive assessment.

Specific Comments
Does the first paragraph in the results section (Lines 189-197) fit better in the methods section?

→ Thanks for the suggestion. We agree with you. We have moved the first paragraph in the results to the algorithms subsection in the methods section (**Lines 194-202, 215-221**).

It is mentioned that the model offers "faster computation speed and a wider valid time scope" (Line 341) but the specific time scope is not detailed. In lines 357-358 it says the wider valid time scope spans approximately one full solar cycle. This would be useful to move to when it is first discussed in line 341. However, it isn't clear what the actual time range of the application and its predictability is – what calendar years is the model able to predict Es layers in? The time scope is wider than what? The SELF-ANN GitHub page is referenced but the time scope isn't clear from the readme there either.

→ Thank you for highlighting this issue. We concede that the manuscript did not fully clarify the specific effective time range for the simplified version of the SELF-ANN tool. Predictive experiments were conducted with the simplified model across different 12-year interval. We have modified the sentence structure and incorporated relevant descriptions (**Fig. 8; Lines 387-393, 395-396, 410-414; Captions Fig. 8**). Additional information has also been included in the GitHub readme (https://github.com/RuleNHao/SELF-ANN).

→ To enhance computational speed, we developed a simplified version of SELF-ANN, which requires only time and spatial parameters to predict the corresponding S4 index with response times at the scale of seconds. The training data for this tool encompassed the COSMIC data from 2007 to 2018, a span covering one full solar cycle. Typically, the model demonstrates superior performance within the scope of the training dataset. To evaluate performance beyond the training data, we varied the input years in the model to obtain corresponding predictions, subsequently contrasting them with observational data.

→ More specifically, we evaluated the model's performance over different 12-year cycles by varying the input year intervals, ranging sequentially from 1987 to 1998, 1988 to 1999, up to 2027 to 2038. We observed that the model achieved its peak performance within the training period from 2007 to 2018 (correlation coefficient was 0.804), with a subsequent decline in performance as the time frame extended. Ultimately, we selected 2002 to 2025 as the effective time range for the simplified model. Evaluation metrics within this range remained within acceptable bounds: for 2002-2013, MAE was 0.323 and RMSE stood at 0.484, while for 2014-2025, the MAE was 0.245 and RMSE was 0.378.

→ The revised Fig. 8 below presents an evaluation of the predictive performance of the simplified SELF-ANN,

including the specific effective time range in Fig. 8a. Figure 8b-8c remain unchanged. Figure 8d displays the statistical results, with blue arrows denoting the range of training data, and bidirectional red arrows indicating the effective time frame of the tool.

[Figure]

Technical Corrections
(1) Line 1: Consider rephrasing to "Ionospheric sporadic E layers (Es) are intense plasma irregularities between 80 and 130 km in altitude, which are generally unpredictable".

→ Corrected (**Lines 1-3**).

(2) Line 26: It may be useful to elaborate/break this down further into the different formation mechanisms, splitting the references up.

→ We agree with you, corrected (**Lines 29-37**).

→ In the revised manuscript, we primarily detailed the formation mechanisms of the Es layer at different latitudes. We briefly outlined the wind shear formation mechanism of temperate or mid-latitude Es, the equatorial electrojet mechanism at equatorial or low-latitude Es, and the auroral particle precipitation factors at auroral or high-latitude Es.

(3) Line 88: "We has" => "we have".

→ Corrected (**Lines 100-102**).

(4) Figure 3: Consider revising colour scheme to improve contrast between areas of high/low intensity. Do the boxes need to be shaded? Just a red outline may be clearer.

→ Thanks for your suggestion. The caption and description of Fig. 3 have been modified (**Fig 3; Lines 263; Captions Fig. 3**).

→ To enhance contrast, we modified the color scheme and the display range of the color bar. Concurrently, shadows on the boxes were removed and replaced with numerical identifiers, as shown below. Original image on the left, modified image on the right.

[Figure]

(5) Lines 284-285: Consider rephrasing to "due to the ions failing to converge vertically" or "since the ions fail to converge vertically".

→ Corrected (**Lines 325-327**). We have rephrased to "due to the ions failing to converge vertically".

(6) Line 297: "in this study, as depicted in" => "in this study is depicted in"

→ Corrected (**Lines 339-341**).

(7) Line 321-324: Consider removing "trained on COSMIC satellite RO data" and putting foEs in brackets to aid

readability.

→ Thanks for your suggestion. Corrected (**Lines 365-370**).

(8) Line 325: Consider removing "square".

→ Corrected (**Lines 370-373**).

(9) Line392: "In sum" => "In summary".

→ Corrected (**Lines 449-452**).